# Theories used to explain care-leavers' journey out of care: A scoping review

Adrian D. van Breda ⓘ *, Sasambal Reuben

Department of Social Work & Community Development, University of Johannesburg, Johannesburg, South Africa

* avanbreda@uj.ac.za

## Abstract

This study was motivated by Mike Stein's 2006 critique of care-leaving research as reflecting a 'poverty of theory'. A scoping review of care-leaving journal articles was conducted for the nine-year period from 2015 to 2023. 252 articles met the inclusion criteria, including that theory must be explicitly mentioned. 133 theories were used across these publications, with resilience theory being used in 24% of articles, life course in 10%, emerging adulthood and attachment in 8% each, social capital in 6% and ecological theory in 4%. Over half of the publications were driven by theory (from conceptualisation, through research design, to interpretation of findings), while a fifth were informed in parts by theory and a quarter alluded only briefly to theory. Three quarters of the articles utilised theory to formulate practice recommendations for care-leaving services. Only two theories could be identified that were constructed to explain care-leaving. Although most theories considered the care-leaver within their social environment, there was little use of structural, systemic, critical and rights-oriented theories. The study concludes that Stein's original concern has been somewhat addressed over the past several years. Other studies find that around a quarter to half of care-leaving publications use theory. This study confirms that, of those studies that make at least some use of theory, most weave theory firmly into the study and mobilise that theory to make recommendations for practice. Nevertheless, research should become yet more theory-driven, contextual, systemic, rights-oriented and critical theories should be used more frequently, and more needs to be done to build theory for care-leaving.

## Introduction

Children and adolescents, whose families are unable to provide them with adequate care, come to the attention of child welfare services and, in some cases, are placed in alternative care [1,2], also known as out-of-home care [3] or foster care [4]. Alternative care may range from a short-term emergency placement to a

**Data availability statement:** All relevant data are within the paper and its Supporting Information files. As this is a scoping review of other research, the paper contains methodology and reporting based on the 'Preferred Reporting Items for Systematic Reviews and Meta-Analysis'. The paper includes database and search strategies and findings, which support replication of the article. There is no ethical or legal restrictions relating to this paper and the submission contains all raw data required to replicate the results of the study.

**Funding:** The author(s) received no specific funding for this work.

**Competing interests:** The authors have declared that no competing interests exist.

permanent placement and may include foster care (with a relative or non-related foster carer), residential or group home care (including children's homes that may be small family-like homes or larger institutional settings) or adoption. The principle preference is for children to remain in their family of origin (with their parents or other family members) or another family (non-related foster care), rather than in residential placements [1].

While in principle, placements out of the family should be as brief as possible [1], in practice, alternative care placements often continue for two or more years, often until the young person reaches the legal age at which they must leave care (usually at age 18), though care can often be extended by a few additional years [5,6]. At this stage, the young person typically must transition out of the child protection or child welfare system towards independent living. This transition is known as 'leaving care' or 'care-leaving', and the young people leaving care as 'care-leavers' [7].

There has been a rapidly growing interest in care-leaving and care-leaver outcomes over the past several years [8]. This is most prominent in North America, Australia and Europe, but also now in many countries in Africa and Asia [8–13].

The interest in care-leaving results from several factors, particularly the body of research that shows the comparatively negative outcomes of care-leavers compared with other young people transitioning out of families [14–17]. While some have argued that it is alternative care itself that produces these outcomes [18], others argue that the children who come into the alternative care system are already at higher levels of vulnerability than children who do not [19,20].

Notwithstanding the growing body of literature on the transition out of care and post-care outcomes or markers, Stein, in his seminal 2006 paper [21], lamented the poverty of theory underpinning this research, which had tended to be empirical and descriptive. He writes, "Although there is a growing body of international empirical work on young people aging out of care, very few of these studies have been informed by theoretical approaches" [21]. His paper proposes three theories that could be useful in informing care-leaving research – attachment, focal and resilience theories – within an overall framework of social exclusion.

Many authors have heeded Stein's call; much of the current care-leaving literature appears to have a stronger theoretical base and some work is being done to develop theories of care-leaving. For example, the first part of the edited book by Mann-Feder and Goyette [8] is termed 'theoretical perspectives', and covers developmental [22], network [23], resilience [24] and human rights [25] theories. A recent edited volume [10] foregrounds theories and/or theoretical constructs like stability and instability, the interpersonal-psychological theory of suicide, social capital and resilience.

However, while there appears to be a growth in theory-informed and theory-building research, there has not been a review of these theories, to determine what theories are being used and how they are used to explain the care-leaving transition. Such a review would be helpful in mapping out the range of theories that inform the journey out of care. This could, in turn, strengthen the development of theory-informed interventions for care-leavers.

The aim of this paper, therefore, is to report the findings of a scoping literature review of recent care-leaving literature to answer the following question: What theories are used or being generated to explain the transition out of care towards young adulthood?

The following sub-questions are asked:

1. What are the most frequently used theories to explain care-leaving?

2. To what extent does theory inform care-leaving research?

3. To what extent do theories inform practice recommendations?

4. What bespoke care-leaving theories have been constructed?

The following section sets out the methodology used in conducting this review. The reporting of the review was informed by the Preferred Reporting Items for Systematic Review and Meta-Analysis (PRISMA) Statement and checklist [26] (S1 Checklist). We did not register a scoping review protocol for this study. Thereafter, the findings of each of the research questions are set out. The findings are finally discussed, conclusions drawn about the state of theory in care-leaving research and recommendations made for the way forward.

## Methodology

Munn, Peters [27] state that a scoping review is differentiated from a regular literature review in that it "includes exhaustive search for information", aims to be "transparent and reproducible" and includes "multiple reviewers" to improve reliability. Scoping reviews act as a way for gaps in the current literature to be recognised, allowing further research to be conducted. The authors conducted a scoping literature review of studies that used one or more theories to explain care-leavers' journey out of care. To the authors' knowledge, this is the first scoping literature review of the theories that are used to explain care-leavers' journey out of care.

### Search strategy

A scoping review search typically includes a selection of information sources inclusive of electronic databases. For the purposes of this study, a selection of electronic databases was searched. Searches were conducted in February 2024. An initial meeting was held with an independent expert, who was well versed in the field of literature searches and who guided and advised on the manner to proceed with the initial search. Thereafter a comprehensive search of literature was undertaken within the following databases: EBSCOhost, Project Muse, ProQuest Central, ProQuest Sociology, Sage, Science Direct, Scopus, SpringerLink, Taylor & Francis and Wiley Online.

The following search terms were used for all databases

"Leaving care" AND theor*

An initial search using a wider range of terms could not be accommodated by several of the databases and was thus abandoned in favour of these simpler terms. In addition, an initial search of the full texts of the publications in the databases generated tens of thousands of hits, with very few relevant ones; thus, the search terms were restricted to title, keywords and abstract.

### Inclusion and exclusion criteria

The inclusion and exclusion criteria were the following:

1) All studies had to have a publication date (including an early view or online date) within the nine-year period of 2015–2023;

2) Publications had to be written in English, for practical reasons and to avoid the cost of translation, but were not limited to any location, country or region;

3) We excluded books, book chapters, book reviews, conference proceedings, editorials, policy guidelines and grey literature;

4) Journal articles had to focus on young people preparing to leave care, young people in the process of leaving care or people who had left care (whether recently or long ago); and

5) The articles had to include (at least in passing) theory used or developed to explain care-leavers' transition out of care.

The year 2015 was selected as the start date for the search because the authors opted to use Van Breda's South African grounded theory study [28] as a point of reference for this study. We regard his paper as a first attempt in South Africa to develop a care-leaving theory, and thus pegged it as our starting point for the review. To ensure trustworthiness of the data, the EBSCOhost database was used as a pilot study, and we thereafter carried out the same search on the other databases. According to Shamseer, Moher [29], piloting is seen as strategic in the selection process "to minimize errors", thereby allowing for further trustworthiness in the overall study.

### Screening steps

The authors worked co-operatively throughout the search process, examining and reviewing each other's work. This was important to ensure trustworthiness of the study.

1) We conducted the pilot search of EBSCOhost together and deliberated over the search terms to ensure they balanced inclusivity and manageability.

2) We then conducted the searches of all the remaining databases together, exporting all data to .ris or .enw files. These were imported into EndNote, which we used as our data management platform.

3) The first author used EndNote's "Find Duplicates" function to identity and delete duplicate articles. This author also manually scanned all the remaining articles to identify and delete duplicates.

4) The remaining articles were divided in half, with each author screening their half of the articles for the inclusion/ exclusion criteria. This review was based primarily on the title and abstract, and where necessary, the full text was downloaded to finalise the decision. When there was uncertainty about excluding an article, we consulted with each other. We then screened all of each other's work in full to ensure consensus. A list of reasons for excluding articles was kept.

5) The first author downloaded and attached the full texts of each article into the database of screened publications. These were again divided in half, with each author reading and screening half the manuscripts. This led, in some cases, to additional articles being dropped due to not meeting the theory criterion (for example, articles that mentioned using grounded theory methods but that didn't develop a theory of leaving care). The authors consulted each other on all such exclusions.

6) The first author 'cleaned up' the EndNote database, ensuring correct formatting of titles, sequencing of author first and last names, ensuring DOI numbers, etc.

7) The final collection of papers included in the scoping review totalled 252 records.

   The PRISMA flow diagram of the study selection process is displayed in Fig 1.

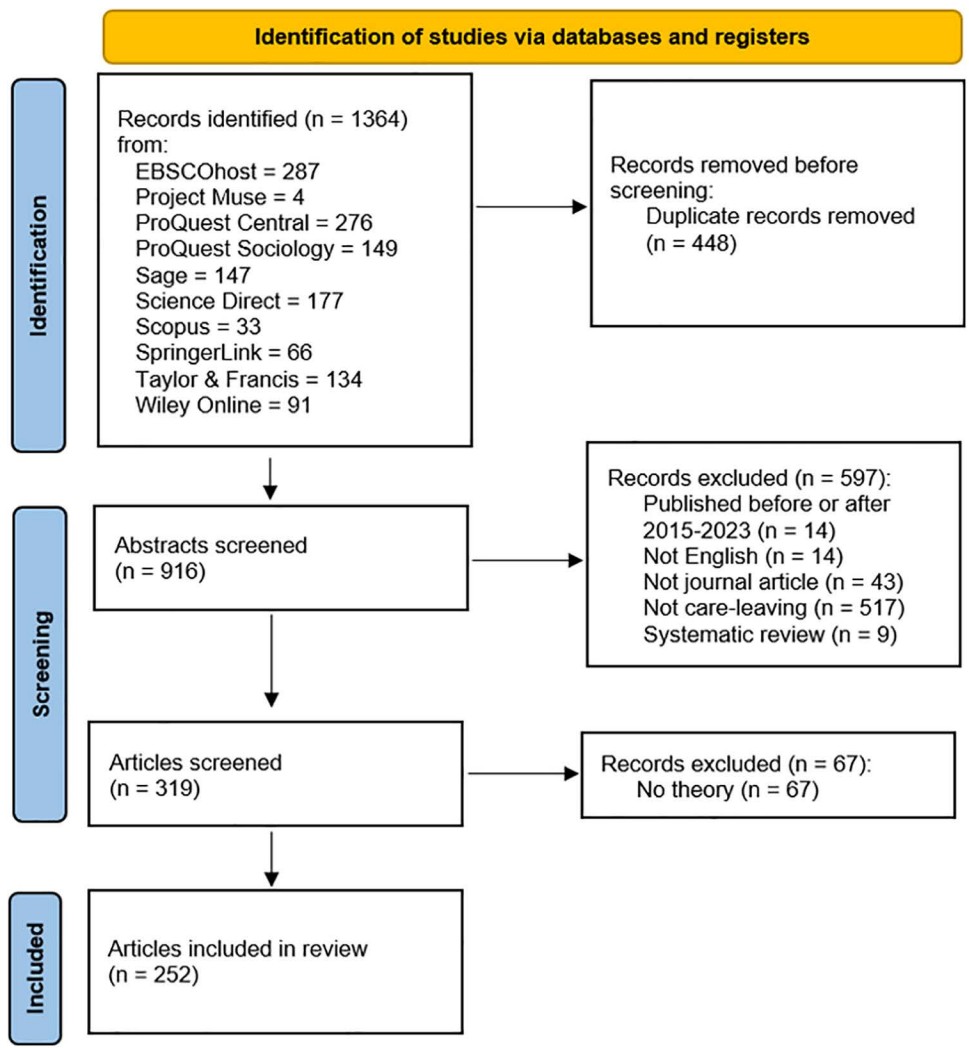

**Fig 1. PRISMA Flow Diagram.**

## Data analysis

The authors exported the references from EndNote into a Word table format and again divided the final collection of publications between them. Each article was scrutinised to identity the theory or theories that were used in the article. When an article mentioned more than one theory, the entry of that article was duplicated, so that each theory per article could be analysed. We then analysed how the theory was used to explain the care-leaving process and whether there were any practical applications emerging from the theory (not from the findings per se), writing our notes into the data extraction table (see S1 Table).

Through this process of analysing the data we began to recognise that some publications made extensive use of theory while others only passing use. The authors' conversations in relation to the articles led to the development of a typology of the role or place of theory in the publications. An iterative process of formulating the typology and then testing it against articles led us to a three-fold typology, viz. theory-driven studies, theory-informed studies and theory-alluded studies, which will be discussed in the findings.

In our analysis, we worked both to quantify the theories used and how they are used, and to qualitatively illustrate the use of theory in explaining care-leaving processes.

## Findings

A total of 252 articles met the screening criteria and comprise the dataset for this study. Some articles appear more than once in the dataset, because multiple theories in one article created multiple rows in the table, which we refer to as theory-article pairs. For example, the 2022 article by Achdut et al. [30] refers to both human capital and life course theory, constituting two theory-article pairs. In total, the review comprises 342 theory-article pairs. The distribution of articles and theory-article pairs per year is depicted in Fig 2. The graph shows no trend in either the number of theory-informed care-leaving publications per year over the nine-year period – articles range from 14 to 42 – nor the number of theory-article pairs – which ranged from 17 to 56. The average theories per article per year range from 1.21 to 1.56 across the nine years.

## Theories used

All articles in this review used at least one theory, as this was core to the sampling criteria. On average, each article used 1.36 theories. In total, 133 theories were identified. Ninety-four (70.7%) of these theories were used in only one article and a further 17 (12.8%) theories were used in only two articles – these will be listed, but not discussed. The remaining 22 (16.5%) theories were each used in three to 64 articles and form the focus of analysis. S1 Table contains details of all articles and theories.

The 94 theories that were used only once are: Adultification [31], Anchors for deliberation [32], Anti-Black Racism [31], Arrested adulthood [33], Behaviour Determinants Intervention [34], Care [35], Citizenship [36], Communities of Practice (CoP) [37] , Conservation of resources [38], Constellations of participation [39], Continuum [40], Convoy model [41], Critical social theory of youth empowerment [42], Cultural capital [43], Cultural connectedness [44], Cumulative advantages and disadvantages [45], Decoloniality [46], Delayed adulthood [33], Dialectical critical realism [47], Drama therapy [48], Dramaturgical approach [49], Dyadic responses to trauma theory [50], Early maladaptive schemas [51], Emotional intelligence [52], Emotional-management model [53], Epistemic injustice [54], Existential well-being [55], Experience [56], Explanatory modelling [57], Family practices approach [45], Family systems [58], Family theory [59], Feminism [60], Feminist pathways [61], Future perspectives [62], General strains [63], Habitus [64], Home [65], Hope [66], Human Capability

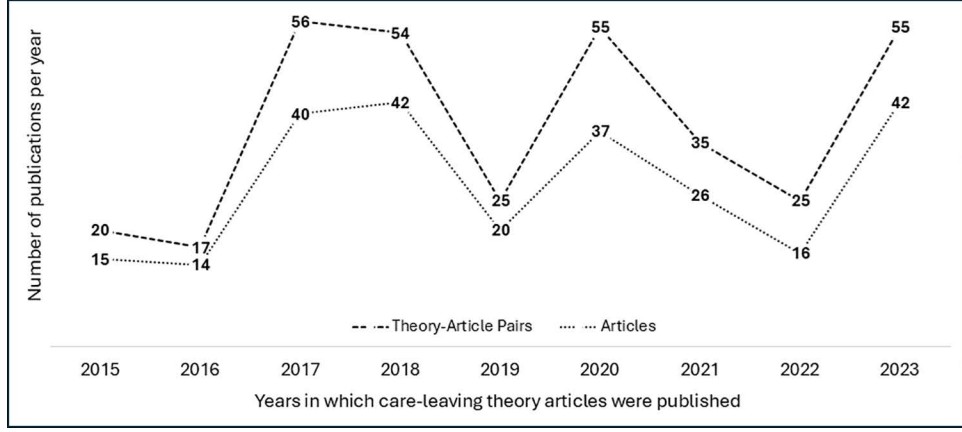

**Fig 2. Theory-Informed Care-leaving Publications per Year.**

Approach [67], Human capital [30], Human development [68], Identity capital [69], Individualisation [70], Institutional ethnography [71], Institutionalised vulnerability [72], Kinning [73], Learned helplessness [74], Lifespan theory of control [75], Marxism [76], Mentoring [77], Minority stress [78], Multilevel model of college access [79], Narrative-based model of identity [80], Neoliberalism [81], Network [82], Nordic welfare model [83], Objectification [84], Occupational justice [85], Open systems [86], Person-centred [87], Person-in-Environment (PIE) [88], Personal construct [89], Planning norms [32], Positioning [90], Reciprocity [91], Relational autonomy [92], Resistance [35], Resources [93], Sanctuary Model [94], Self as social construction [95], Self-efficacy [96], Sense of community [97], Shared deliberations [32], Social citizenship [98], Social constructionism [99], Social control [100], Social identity [101], Social justice [102], Social model of disability [103], Social reproduction [104], Social sustainability [105], Stigma [106], Street-level bureaucracy [107], Symbolic interaction-ism [108], Teaching-family [109], Theory of change [110], Threatened identity [106], Transition [111], Transnational [112], Trauma [113], Trust, risk and uncertainty [114], Turning points [115], and Youth mentoring [78].

The 17 theories that were used only twice are: Ambiguous loss [116,117], Belonging [118,119], Critical disability [71,120], Empowerment [85,121], Ethic of care [122,123], Family [124,125], Hierarchy of needs [126,127], Homeostasis [128,129], Institutional logics [120,130], Motivation [131,132], Narrative therapy [48,133], Precarity [134,135], Psychoso-cial [136,137], Social exclusion [103,138], Social learning [93,139], Social network [140,141], and Ubuntu [116,142].

Table 1 lists the 22 theories that were used in three or more articles in descending order of frequency. Percentages in the frequency column refer to the percentage of the 252 articles in which each theory is used. Together, these 22 theories account for 63% of all the theory-article pairs (214 out of 342). The remaining columns (Driven, Informed and Alluded) will be discussed later in the article.

It should be noted that theories are sometimes difficult to label or frame, because some theories and theoretical con-structs overlap and others form components of larger theories. For example, life course theory features frequently, but comprises several sub-themes, such as linked lives. Some articles [216] centred on linked lives and not the other compo-nents of life course, while other articles focus on life course, giving limited attention to linked lives [45]. In such cases, we endeavoured to indicate this by coding the theory at multiple levels, e.g., "Life course – linked lives" in the case of Brady & Gilligan [216], and simply "Life course" in the case of Adeboye et al. [45].

Resilience theory [279,280] is the most prominent theory used to explain care-leaving. It appears in a quarter of the articles. While resilience theory used to be highly individualistic and could be used to infer that care-leavers were entirely responsible for their own care-leaving journey, a socioecological approach to resilience is increasingly used [237, 250, 281], attending to personal, relational and systemic or structural resilience enablers. Resilience theory has the advantage of helping to identify the kinds of resilience processes that facilitate a smoother transition out of care.

Life course theory [282] is the second most prevalent theory in care-leaving research, appearing in a tenth of articles. It helps to explain how one's journey through life is influenced by personal and environmental factors, and thus connects with cognate concepts like agency, linked lives and social capital. It was seldom used in the past – only eight articles in the five years from 2015 to 2019 – but has become prominent this decade – 15 articles in the three years from 2020 to 2023 [204, 217].

Emerging adulthood [283] is the third most prevalent theory (8%). It refers to a stage of human development corre-sponding roughly to ages 18–29, which is the age range in which most care-leaving takes place and is thus the age range of most care-leaving research. Although emerging adulthood theory has several core concepts (as does life course theory), much of the care-leaving research that uses emerging adulthood uses the term primarily to refer to an age range [74, 172, 183].

Attachment theory [284] is a fourth theory shaping the theorisation of care-leaving (8%). This is understandable given that being placed in care frequently results from ruptures in children's primary attachment relationships and the removal from the family and placement into care, while the later aging out of care are additional disruptions of attachment. Notions

**Table 1. Prominent theories for care-leaving.**

| Theory | Citations | Frequency | Driven | Informed | Alluded |
|---|---|---|---|---|---|
| Agency | [108,143–147] | 6 (2%) | 4 (67%) | 2 (33%) | 0 (0%) |
| Attachment | [28,41,50,93,148–162] | 19 (8%) | 8 (42%) | 4 (21%) | 7 (37%) |
| Developmental social welfare | [14,147,163] | 3 (1%) | 0 (0%) | 0 (0%) | 3 (100%) |
| Ecological | [50,147,164–171] | 10 (4%) | 5 (50%) | 4 (40%) | 1 (10% |
| Emerging adulthood | [5,33,46,65,104,158,172–186] | 21 (8%) | 8 (38%) | 3 (14%) | 10 (48%) |
| Focal | [5,187–189] | 4 (2%) | 1 (25%) | 1 (25%) | 2 (50%) |
| Future orientation | [62,127,178,179] | 4 (2%) | 3 (75%) | 0 (0%) | 1 (25%) |
| Identity | [190–192] | 3 (1%) | 2 (67%) | 1 (33%) | 0 (0%) |
| Interdependence | [36,190,193] | 3 (1%) | 3 (100%) | 0 (0%) | 0 (0%) |
| Intersectionality | [70,194,195] | 3 (1%) | 2 (67%) | 0 (0%) | 1 (33%) |
| Journey towards independent living | [28,196,197] | 3 (1%) | 3 (100%) | 0 (0%) | 0 (0%) |
| Life course | [30, 45, 69, 184, 198–217] | 24(10%) | 18 (75%) | 3 (13%) | 3 (13%) |
| Liminality | [134,135,198] | 3 (1%) | 2 (67%) | 0 (0%) | 1 (33%) |
| Possible selves | [173,218,219] | 3 (1%) | 2 (67%) | 1 (33%) | 0 (0%) |
| Recognition | [134,135,220] | 3 (1%) | 3 (100%) | 0 (0%) | 0 (0%) |
| Resilience | [28, 35, 38, 41, 67, 69, 72, 90, 96, 132, 148, 151, 153, 158, 161, 163, 170, 173, 178, 180, 189, 196, 197, 206, 212, 214, 37,221–257] | 64 (25%) | 29 (45%) | 16 (25%) | 19 (30%) |
| Self-determination | [96,110,116,253,258–261] | 8 (3%) | 5 (63%) | 3 (38%) | 0 (0%) |
| Social capital | [28,63,206,246,256,262–271] | 15 (6%) | 8 (53%) | 3 (20%) | 4 (27%) |
| Social support | [190,272–274] | 4 (2%) | 3 (75%) | 1 (25%) | 0 (0%) |
| Stein's groups | [226,231,275] | 3 (1%) | 1 (33%) | 1 (33%) | 1 (33%) |
| Strength-based | [195,239,244,250,276] | 5 (2%) | 1 (20%) | 1 (20%) | 3 (60%) |
| Sustainable livelihoods | [60,277,278] | 3 (1%) | 2 (67%) | 1 (33%) | 0 (0%) |

of relational permanence, belonging and placement stability are informed by attachment theory, and are considered key to the well-being of care-leavers [285]. Attachment theory is thus important for care-leaving research [151,153,156].

Social capital, with its concepts of bonding, bridging and linking [286], is the fifth most prominent theory used in recent care-leaving publications (6%). Social capital intersects with resilience and life course theories, making it a rich connector for theorisation. All three categories of social capital are shown to be helpful for care-leavers in different ways and thus an important construct for this population [256, 266, 267].

The last theory to be presented here was found in 10 articles (4%): ecological or social ecological theory [287]. This theory aligns with the person-in-environment principle that is core to social work's identity [288,289]. It links to resilience and life course theories, and also to attachment and social capital, because of its focus on the social environment. It thus informs several care-leaving studies [170,171,290].

It is not possible in this article to review all 133 theories that appear in care-leaving publications over the past nine years. It is apparent, though, that there is a wide spectrum of theories that could be helpful for theorising young people's experiences of transitioning out of care into young adulthood, and that some of those that are not in the six we have briefly discussed here may be pregnant with possibilities, as seen in life course theory that has rapidly risen into prominence in the past few years. Other theories that are marginal in care-leaving research, such as possible selves, ethic of care, liminality, institutional logics and anti-Black racism, could be found by researchers to be valuable to shed fresh light on the leaving care journey.

## Theory-informed care-leaving publications

This study did not aim to determine what percentage of care-leaving publications incorporated theory. Instead, it aimed to understand how theory was used in those papers that did include theory. Nevertheless, in the process of gathering data, several topic-focused scoping reviews provided information on the percentage of papers that included a theoretical framework. Gahagan et al.'s [291] scoping review of publications on post-secondary education among care-leavers found that only 13 out of 58 studies (22%) published between 1997 and 2022 stated an explicit theoretical framework. Eight of these used social-ecology, three life course, and one each intersectionality, feminist and phenomenological.

Another scoping review, conducted by Stubbs, Baidawi [292], focused on care-leavers' experience of informal support. Fifty-eight papers, published from 2013 to 2020, were included in the review, thus a more recent sample of publications than Gahagan et al.'s study above. The authors reported that 57% of the studies included theoretical explanations, and that resilience theory was the most prevalent theory (one quarter of publications). Thereafter, 14% used social capital theory, 12% emerging adult theory, and 9% attachment theory, comparable to the results of this study (see Table 1) even though their study had a narrower focus on informal support for care-leavers than our focus on care-leavers per se.

Hodgson, Cordier [293] developed a matrix guide for analysing qualitative care-leaving data. As part of their study, they mapped theories that inform care-leaving research into a graphic (Fig 3). Larger circles imply more studies using those theories. The authors note the emphasis on individual or personal theories, rather than macro, structural and sociological theories, and point to right-based theories as emergent.

## Use of theories to explain care-leaving

Through the review of the use of theories of the selected articles, we identified three main ways in which theories were used or the place of theory in the studies. We termed these as theory-driven, theory-informed and theory-alluded studies. All three of these categories did use theory, but in different ways. Over half of the papers were theory driven (53%), a fifth were theory informed (21%) and a quarter were theory alluded (26%).

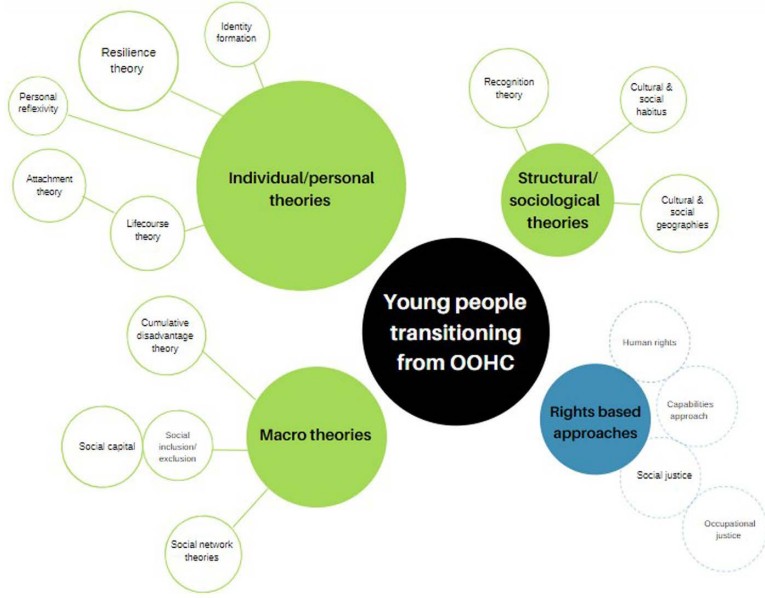

**Fig 3. Theories of Leaving Care.**

First, **theory-driven studies** were those in which theory was introduced prior to the methodology section, thereby shaping the study design. Many of these studies constructed the research question in theoretical terms, e.g., to apply a theory to the data or to test a theory's relevance in practice. In most of these cases, the theory was woven throughout the study, from introduction to conclusion.

We found that over half (n = 197, 58%) of theory-article pairs were theory driven.

For example, Barratt, Appleton [143] build on a previous related study [146] to explore the relevance of Archer's concept of internal conversations within her broader theory of agency. Archer's theory is introduced and explicated in the opening pages of the study. The study aim is framed within the theory, viz. "Our aim in this small-scale research project was to use the internal conversations model to explore how care leavers come to understand, interpret and then act (or not act) in relation to their current context and past experiences" [143]. The research methodology explicitly focuses on these internal conversations. For example, the first interview with care-leavers was an "in-depth, semi-structured, interview which focused on their internal conversations" and the interview scheduled "was informed by Archer (2003, 161-162)" [143].

The eight themes that emerged from Barratt et al.'s [143] study are similarly framed within Archer's theory, for example, "The fourth theme, internal conversations as unhelpful or ineffectual, highlights that for some care leavers internal conversations could be experienced as unhelpful or even harmful", and "The last three themes all relate to the overarching complexity of social relationships for this group and how this was expressed and worked through in internal conversation". The discussion reflects on the findings and draw on Archer's theory and other studies on internal conversations. For example, "we found that the internal conversations and meaning-making of the participants are powerfully shaped by their experiences of trauma and disappointment, which may inhibit their sense of themselves as active agents in the world" [143].

The theory drivenness of this study is crudely evident in the fact that the term 'internal conversation' appears 61 times in the article. A comparable example is the study of Bennwik and Oterholm [36] on policy values related to support for care-leavers with disabilities. A key theory informing this study is citizenship, even though the term does not appear in the title and only once near the end of the abstract. However, the term is used 33 times in the article, appearing on every page except the first page.

Bennwik and Oterholm [36] introduce the term 'citizenship' on the second page in a section titled 'central theoretical perspectives'. On the following page they theorise the construct, stating, "Citizenship as a concept has changed and developed over time, but fundamentally it deals with questions of membership in society based upon certain rights and obligations." They refer to Nussbaum in asserting that "all human beings are citizens, independently of their contributions to society".

The study was conducted through a document analysis, which included a search for the term 'citizenship'. They extended the notion of citizenship to inclusive citizenship, that de-emphasised the contribution a citizen makes to society. For care-leavers with disability, this meant decentring the notion of 'independent adulthood' and centring on the notion of 'inclusive citizenship'.

Second, **theory-informed studies** were those in which theory was either introduced early in the paper, but not pulled through into the methodology, findings and discussion, or introduced mainly in the discussion section, as a way of retrospectively illuminating and explaining the research findings. The study may have been driven primarily by empirical concerns and not framed within a theory. But once the data was in hand, the researchers identified one or more theories to help make sense of and theoretically contextualise the study findings.

We found that a fifth (n = 67, 20%) of articles were theory informed.

Some theory-informed studies introduce theoretical concepts in the opening pages of the article, but do not pull them through to the methodology, findings and conclusions. For example, Goyette, Blanchet [150] examined the "role of placement instability on employment and educational outcomes among adolescents leaving care". On the second page, in a literature review, they write an intensive paragraph on attachment theory, which they argue helps to explain "the negative

outcomes associated with placement instability" [150], citing Bowlby and numerous other authors. After page 2, attachment is never mentioned again until the reference list. Although attachment theory apparently informed the conceptualisation of the study, in relation to placement instability, it did not drive the study in the way described in the previous two theory-driven studies.

Other theory-informed studies brought theory in towards the end to interpret findings. For example, Mupaku, Van Breda [189] studied the experience of care-leavers with disabilities transitioning out of residential care during the COVID-19 pandemic. The study is not theoretically framed up front. But in the discussion of the findings, social-ecological resilience is introduced to explain how COVID-19 led to an atrophy of social-ecological resources – family, friends, work, education – which lowered the resilience of care-leavers and made them more vulnerable to deterioration in behaviour and mental health.

These authors also draw on focal theory, which is used to explain how the transition out of care during ordinary times entails a pile-up of transitions that overwhelm the capacity of many care-leavers. For care-leavers with disabilities who are transitioning out of care during COVID-19, the pile-up increased substantially, overloading care-leavers with too many transitions at one time.

Finally, in the recommendations, the authors recommend that in line with focal theory, transitional tasks should be spread over a longer period, and in line with social ecological resilience theory, care-leavers and their families need to be buffered with networks of social services.

Third, **theory-alluded studies** were those which mentioned theories or theoretical constructs in passing, but did not link these clearly to the study or its findings. These studies were largely designed to address empirical questions. They did not frame the study or the findings in theoretical terms, but merely alluded to theory in passing.

We found that a quarter (n = 78, 23%) of articles were theory alluded.

An article on patterns of criminal behaviour among care-leavers in South Africa [63] refers to two theories in the literature review: general strains theory and social capital (or bonding capital) theory. Each theoretical construct is explained and applied to criminal activity in a sentence or two. The methodology, however, is not informed by these theories, and they are not used in the discussion or recommendations to explain the findings or make recommendations for policy or practice. These theories are alluded to, but not utilised. Both could possibly have contributed to a richer analysis and interpretation of the data.

An article by Dutta [165] explored the preparation of girls for leaving residential care in India. In the discussion section, the author refers briefly to Bronfenbrenner's ecological theory, identifying three of the levels (micro, meso and macro). In a figure, six factors that affect preparation for leaving care are plotted into these levels. Other than using the levels as headings, there is no further discussion of the findings according to social ecological theory. Thus, this theory is alluded to in this article, but not explicated or applied in any depth.

### Practical application of theories

Most studies on leaving care produce recommendations or implications for practice. Among the 342 theory-article pairs in this review, 81 (24%) did not make recommendations informed by theory. Some may have made recommendations based on the study findings but did not link these recommendations to theory. On the other hand, three quarters of the theory-article pairs (76%) did make recommendations that were, to at least some degree, informed by theory.

For example, Dumont, Lanctôt [218] published a possible selves theory-driven article on the hopes and fears of girls transitioning out of care in Canada. In their implications for practice, they recommended that social workers and other care workers should facilitate young people to reflect on their hoped-for possible selves, and to develop these into viable future selves. On the other hand, the girls needed support in avoiding their feared possible selves. They recommended also that they explore the actual pathways to achieving these selves. And they encouraged workers to find role models who aligned with each girl's possible selves, to help strengthen and actualise these possible selves into actual selves.

Berejena Mhongera [60] utilised sustainable livelihoods theory in her study of adolescent girls transitioning from care in Zimbabwe. To build livelihoods in the face of the gender-specific challenges girls face leaving care, she recommends including girls in participative decision making at both personal and strategic levels to develop competencies required in adulthood. She recommended fostering connections between societal systems to facilitate their participation in these systems and to develop self-efficacy and the capacity to advocate. Access to a range of assets should be facilitated, cognisant of the various ways girls are excluded from assets in a patriarchal context, so that they can draw on these assets after leaving care.

Hollingworth and Jackson [187] reanalysed data from two previous studies in the UK and Europe utilising focal theory, to determine if focal theory "would have helped predict their educational progression or otherwise" [187]. They found that, indeed, care-leavers were often overwhelmed with the pileup of challenges in rapid succession. The authors recommend that care-leavers be given more latitude in planning their own pathways out of care, rather than being disengaged on their 18th birthday. This would allow care-leavers to separate and sequence the numerous tasks and challenges they must navigate in leaving care. The authors argue that practitioners need to understand focal theory and its implications for transitional planning, which they suggest could be done in half a day. Such training would make practitioners more mindful of the pileup of challenges and more confident to find ways and take time to space these challenges out. Finally, they indicate that care-leavers themselves would also benefit from an understanding of focal theory, so that they could identify periods of pileup and negotiate more spacing for themselves.

Finally, we draw on six studies that use self-determination theory – four as theory-driven [96,258–260] and two as theory-informed studies [116,261]. We are integrating the findings to illustrate how multiple studies, drawing on a common theory, can be synergised to generate integrated practice recommendations. These studies argue that self-determination is a relational construct, rather than a personal one, and that both the capacity or opportunity to self-determine and processes of self-determination occur in socio-cultural contexts. Females raised in ultraorthodox religious communities, for example, may not be permitted to exercise self-determination, thus they need to learn to recognise and express their needs in partnership with a supportive other. They also need to be given life experiences that build optimism for greater self-determination. Thus, self-determination – whether and how it is exercised – must be considered within each person's cultural contexts. All care-leavers need time to rest, reflect and think to enable the capacity for self-determination. There is a need for opportunities to build trusting and emotionally safe relationships with significant others, so that self-determination of their futures can be negotiated in relational contexts. To this end, working towards interdependence, rather than independence as the goal for leaving care is necessary. Interdependence emphasises the mutual supportive interconnections between people that enable them to flourish [193]. Mentoring is a potentially valuable practice for supporting self-determination, by building diverse voices into a care-leaver's network of relationships. In combination, these interventions will foster relatedness, competence and autonomy, contributing to effective self-determination.

## Theories developed specifically for care-leaving

Only two studies could be found that appeared to construct a theory of or for care-leaving. Several studies were sampled because the term 'grounded theory' was used, but excluded from the review because there was no evidence of a 'theory' being formulated. There are various types of theories [294], the most relevant of which for this paper was explanatory theories, i.e., theories that helped to explain the care-leaving journey. Five studies that used grounded theory methods were included in the review, but did not produce what we could recognise as 'a theory of care-leaving' [65,183,202,205,233].

A grounded theory study that did appear to meet the requirements of an explanatory theory was Van Breda's model of the 'journey towards independent living' [28,196,197], which comprises four interconnected psychosocial processes: striving for authentic belonging, networking people for goal attainment, contextual responsiveness and building hopeful and tenacious self-confidence. These four processes are constrained or held by contextual boundaries and environmental interface [28]. After the original study, conducted only with male care-leavers, the study was replicated with female

participants. The original findings were confirmed [197] and two additional processes specific to female care-leavers were identified, viz. embracing motherhood and taking on responsibilities [196]. Ongoing research on this theory is currently underway, but publications fell outside the timeframe of this study [295].

Another study used Bayesian explanatory network models for building a theory or explanatory model regarding "disruptive behaviour among Finnish care leavers who have been receiving aftercare services for around two years" [57]. A model can be considered 'explanatory' if it presents "the causal connection of the variables and interventions to the outcome variable" [57]. Disruptive behaviour in this study includes regular substance use, threats of violence, problems managing finances, and social interaction patterns [57]. Completing "secondary or vocational education and primary placement at a foster family", as well as the "use of employment office services … were significantly associated with less disruptive behaviour" [57].

## Limitations

Arguably, the biggest limitation of this review was the search term ("Leaving care" AND theor*). Not all research on leaving care uses the term 'leaving care' or 'care leaving' – terms like 'aging out of care' or 'care-experienced people' are used by many scholars. This may have resulted in relevant publications being omitted from the search. We initially developed a comprehensive collection of search terms, but found that most databases could not handle the multiple options or that the search resulted in a massively overinclusive collection of mostly irrelevant information.

Furthermore, the words theory, theorise, theoretical, etc. were not necessarily stated in the article. Publications may have referred to the resilience perspective or strengths approach, for example, and would be missed because of the absence of the word 'theor*'.

Our decision to exclude grey literature, in a context where numerous technical reports on care-leaving are emerging across countries, may have excluded some important theories.

Notwithstanding these limitations and the likelihood that some relevant publications were missed, a large collection of literature was sourced over a nine-year period, which we believe fairly reflects the spread of theories used in care-leaving studies.

All the articles included in this review mentioned theory. We do not know how many articles on care-leaving did not mention theory, and thus are unable to say what percentage of care-leaving studies incorporated a theoretical lens of some sort. For that, an exhaustive review of all care-leaving studies would be required, which was beyond the scope of our study.

A further gap in this review is the non-inclusion of the discipline of the authors. It could have been illuminating to see whether there are differences in theories used by social workers, psychologists, sociologists and others to explain care-leaving.

## Discussion and recommendations

In 2006, Stein raised concerns about the limited use of theory to inform care-leaving research. This clarion call resounded and was taken up by many scholars working in the field of care-leaving. Of the three theories Stein recommended as having potential to explicate the care-leaving process, resilience is the front runner, with attachment also being well utilised. Focal theory lags considerably, though studies using it provide helpful insights into the pile-up of challenges that care-leavers face and have useful implications for practice. By contrast, life course theory, which Stein mentions in passing in the second last paragraph of his paper, is currently the second most frequently used theoretical framework for explaining care-leaving.

The emphasis on theory-informed research does not mean that empirical descriptive research is not useful. It provides useful data on patterns that shape research, practice, policy and resource allocation. However, research that is grounded in theory has the potential to illuminate the experience of leaving care and the design of interventions to facilitate

care-leaving. While this study does not address the question of whether theory use is increasing, two other scoping reviews provide evidence suggestive of an increase in theory-informed care-leaving research over time.

What this study distinctively shows is the diversity in the way theories are utilised in care-leaving research. Over half of the studies that mentioned theory used theory extensively to inform and drive the research questions, study design and interpretation of data, including recommendations for policy and practice. This is encouraging, as such research not only uses theory rigorously, but may also contribute to building and developing those theories.

However, a fifth of studies were theory informed, meaning some use of theory was made, at the front or back of the study, but not pulled all the way through from conceptualisation to recommendations. And a quarter of studies mentioned theories only in passing, which we termed theory-alluded studies. There were studies that were driven primarily by one theory, and that also utilised other theories in an informed or utilised fashion. The studies reviewed utilised, on average 1.4 theories each, suggesting that while some studies focused in on one theory, others may have used two or more theories to weave together a multifaceted theoretical construction of leaving care.

It appears that many, if not most, of the theories used in three or more studies construct care-leaving as a person-in-environment interaction. Some theories, that may be thought of as individual, such as resilience, attachment, agency and focal, are utilised in contextualised ways, foregrounding the interactions between care-leavers and their social environment. At very least, they are located in interpersonal relationships, but frequently they are extended further into interactions with the systems and structures of society.

Other theories, however, have a distinctively contextual and environmental focus, such as social ecological theory, developmental social welfare and sustainable livelihoods, though with the exception of ecological theory these are seldom used. Some other theories lean towards critical theory, such as intersectionality. But others, such as anti-black racism, critical disability, ethic of care, feminism, Marxism, minority stress, precarity and social justice, have been used only once or twice over the past nine years. It will be good to see structural, systemic, critical and rights-oriented theories enjoying greater use in the years to come.

Encouragingly, three quarters of the studies in this review used theory to inform practice recommendations. The extent to which theory informed these recommendations varies from a token nod to theory to a rigorous application of theory into care-leaving practice, but given Stein's [21] concern about "the poverty of theory", even some use of theory to inform practice, and not only data, is encouraging.

What remains sparse are theories specifically developed for care-leaving. Only one viable theory, tested across more than one study, has emerged and it has not gained traction among other scholars. It is, arguably, not necessary to construct new theories if existing theories can be customised or applied to care-leaving. Nevertheless, theory development may prove helpful, not only for care-leavers, but for all young people transitioning from childhood into adulthood.

## Conclusion

In conclusion, we wish to celebrate the care-leaving scholarly community's attention to Stein's seminal call almost 20 years ago. It does appear that care-leaving research is making greater use of theory and when they do, theory is used to drive the research, rather than merely to inform it or to briefly allude to it. This is leading to richer and more experience-near explanatory narratives about the journey of leaving care, which should inform and strengthen care-leaving practice.

We suggest the following areas of future growth: (1) an increase in theory-driven research and a decrease in theory-alluded research, so that theory is used purposefully and rigorously to make sense of the care-leaving experience; (2) an increase in the use of contextual, systemic, rights-oriented and critical theories, particularly given the increasingly fractured society that many care-leavers transition into; and (3) an increase in theory building for care-leaving.

## Supporting information

**S1 Checklist.  PRISMA-ScR checklist.**
(DOCX)

**S1 Table.  Data Extraction Table.**
(DOCX)

## Acknowledgments

Honouring the contribution of care-experienced people: We stand with those who are care-experienced. We acknowledge and respect their stories. We thank them for sharing their expertise and wisdom and their extraordinary contributions to research, policy and practice. Learning from their lived experience and working together towards improving care systems for future generations is a privilege.

We acknowledge the immense contribution of Mike Stein to care-leaving research generally and to the theorisation of care-leaving particularly. He can rightfully be considered the parent of the research field of care-leaving, and is much appreciated.

## Author contributions

**Conceptualization:** Adrian D. van Breda.

**Data curation:** Adrian D. van Breda, Sasambal Reuben.

**Formal analysis:** Adrian D. van Breda, Sasambal Reuben.

**Methodology:** Adrian D. van Breda.

**Supervision:** Adrian D. van Breda.

**Writing – original draft:** Adrian D. van Breda.

**Writing – review & editing:** Sasambal Reuben.

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
