## [Decision Letter · Decision Letter 0]

11 Apr 2025

PONE-D-24-53785Theories used to explain care-leavers’ journey out of care: A scoping review

PLOS ONE

Dear Dr. van Breda,

Thank you for submitting your manuscript to PLOS ONE. After careful consideration, we feel that it has merit but does not fully meet PLOS ONE’s publication criteria as it currently stands. Therefore, we invite you to submit a revised version of the manuscript that addresses the points raised during the review process.

We look forward to receiving your revised manuscript.

Kind regards,

Daryl Higgins, PhD

Academic Editor

PLOS ONE

Journal Requirements:

2. As required by our policy on Data Availability, please ensure your manuscript or supplementary information includes the following:

Reviewers' comments:

Reviewer's Responses to Questions

**Comments to the Author**

1. Is the manuscript technically sound, and do the data support the conclusions?

Reviewer #1: Yes

Reviewer #2: Yes

2. Has the statistical analysis been performed appropriately and rigorously? 

Reviewer #1: N/A

Reviewer #2: I Don't Know

3. Have the authors made all data underlying the findings in their manuscript fully available?

Reviewer #1: Yes

Reviewer #2: Yes

4. Is the manuscript presented in an intelligible fashion and written in standard English?

Reviewer #1: Yes

Reviewer #2: Yes

5. Review Comments to the Author

Reviewer #1: I appreciated the opportunity to read this valuable article. It presents a clear overview of study rationale and aims, fills a major knowledge gap in leaving care research, presents a clear statement of methods and inclusion and exclusion criteria, detailed findings and a sound conclusion. I like the careful definition of ambiguous terms such as resilience, and the explanation of why only six theories used were analysed. I have made a few minor tracked sub-edits throughout the transcript, and posed one brief question. The authors could potentially address/add 2 minor items. One is whether the exclusion of grey literature may have weakened the paper given that there is an increasing number of comparative regional and intra-country reports linked to the emerging INTRAC networks being completed, for example, the 2022 OECD report, Assisting care leavers. Also, on page 26, I wonder if an area for future research could be linking theory to the disciplines of the authors. For example, are social work or psychology or sociology leaving care studies more or less likely to include theory, and if so, which types of theories? Other than that, an excellent paper which will add significantly to leaving care scholarship.

Reviewer #2: 1. This is an easy-to-follow piece of research that answers the questions asked of the study:

1. What are the most frequently used theories to explain care-leaving?

2. To what extent does theory inform care-leaving research?

3. To what extent do theories inform practice recommendations?

4. What bespoke care-leaving theories have been constructed?

The article effectively outlines the study's context and methodology, utilising the established PRISMA reporting system for scoping reviews. The methodology details trustworthiness strategies, inclusion and exclusion criteria, the screening process, sample size, data analysis, and methodological limitations. The way that the article clearly presents this process and justifies the methodological decisions allows for the replication of the study.

2. The statistical analysis presented is relatively minor. As I am not a quantitative researcher, someone with expertise in this area may be better suited to review this specific aspect of the research. However, under the section titled "Findings," specifically at line 186, you mention that the review includes 342 theory-article pairs. It would be beneficial to define what "theory-article pairs" means to enhance clarity. Adding a brief definition, perhaps in parentheses, would make Figure 2 easier to comprehend and help readers better grasp the statistical information in that paragraph.

3. The data is available within the article. While it is essential for transparency to include all the theories examined, could the 94 theories used once, and the 22 theories used twice be moved to an appendix? This might help reduce distractions from the analysis of the other theories that were used multiple times. This is a minor suggestion, and I understand it may not be feasible to have an appendix.

4. Besides the point raised in point 2 about adding a small definition of what theory-article pairs mean, the article is clear and well-written. Essentially, it argues that since Mike Stein's call for care-leaving research to incorporate theory and build a care-leaving theory, researchers have made progress, but there is still much work to be done, particularly in developing care-leaving theory. In addition, the research produced an emerging theoretical framework (analytical typology) that helped analyse how the theory was applied in care-leaving research. This typology is especially valuable for those researching the care-leaving experience.

6. PLOS authors have the option to publish the peer review history of their article (what does this mean? ). If published, this will include your full peer review and any attached files.

**Do you want your identity to be public for this peer review?** For information about this choice, including consent withdrawal, please see our Privacy Policy .

Reviewer #1: No

Reviewer #2: **Yes: ** Dr Jacinta Waugh

---

## [Author Response · Author response to Decision Letter 1]

6 May 2025

PONE-D-24-53785

Theories used to explain care-leavers’ journey out of care: A scoping review

Thank you for the feedback on the manuscript. This is much appreciated.

Revisions required Author responses

Reviewer #1: I appreciated the opportunity to read this valuable article. It presents a clear overview of study rationale and aims, fills a major knowledge gap in leaving care research, presents a clear statement of methods and inclusion and exclusion criteria, detailed findings and a sound conclusion. I like the careful definition of ambiguous terms such as resilience, and the explanation of why only six theories used were analysed. I have made a few minor tracked sub-edits throughout the transcript, and posed one brief question. Many thanks for your positive feedback and for the language corrections and suggestion about interdependence.

Language edits in the manuscript in track changes Thank you for spotting these errors, including the typo in the Children’s Amendment Act in the reference list. We have carefully checked each edit and incorporated them into our revised manuscript. We have not mentioned them individually unless, there was reason for some discussion.

Line 456. You may need to define for the non-specialist reader, what is meant by interdependence. Thank you for this suggestion. We have provided a brief definition: “Interdependence emphasises the mutual supportive interconnections between people that enable them to flourish.” And we have cited Storø’s seminal publication on the concept:

Storø, J. (2018). To manage on one’s own after leaving care? A discussion of the concepts independence versus interdependence. Nordic Social Work Research, 8(sup1), 104-115. https://doi.org/10.1080/2156857X.2018.1463282

Line 458 – remove ‘a’ in “building voices of a difference” This phrase “voices of a difference” comes from the work of Gregory Bateson, but we recognise that this is rather obscure, and have rephrased the sentence as “…by building diverse voices into a care-leaver’s network of relationships”

The authors could potentially address/add 2 minor items. One is whether the exclusion of grey literature may have weakened the paper given that there is an increasing number of comparative regional and intra-country reports linked to the emerging INTRAC networks being completed, for example, the 2022 OECD report, Assisting care leavers. We agree that this is a limitation, and have added it to the section on limitations. While we agree with the limitation, we think that the decision to focus solely on peer-reviewed journal articles was correct, given the large quantity of articles that emerged from the review.

Also, on page 26, I wonder if an area for future research could be linking theory to the disciplines of the authors. For example, are social work or psychology or sociology leaving care studies more or less likely to include theory, and if so, which types of theories? Thank you for this interesting point. We agree that this could have been helpful, or would be helpful in future studies. Although you have phrased it as a recommendation for future research, we have included it in the Limitations section.

Other than that, an excellent paper which will add significantly to leaving care scholarship. Many thanks for your positive assessment of this paper.

Reviewer #2: 1. This is an easy-to-follow piece of research that answers the questions asked of the study:

1. What are the most frequently used theories to explain care-leaving?

2. To what extent does theory inform care-leaving research?

3. To what extent do theories inform practice recommendations?

4. What bespoke care-leaving theories have been constructed?

The article effectively outlines the study's context and methodology, utilising the established PRISMA reporting system for scoping reviews. The methodology details trustworthiness strategies, inclusion and exclusion criteria, the screening process, sample size, data analysis, and methodological limitations. The way that the article clearly presents this process and justifies the methodological decisions allows for the replication of the study. Thank you for these affirming comments.

2. The statistical analysis presented is relatively minor. As I am not a quantitative researcher, someone with expertise in this area may be better suited to review this specific aspect of the research. However, under the section titled "Findings," specifically at line 186, you mention that the review includes 342 theory-article pairs. It would be beneficial to define what "theory-article pairs" means to enhance clarity. Adding a brief definition, perhaps in parentheses, would make Figure 2 easier to comprehend and help readers better grasp the statistical information in that paragraph. Thank you for the prompt on the meaning of ‘theory-article pairs’. We have included an explanation and example of this in the first paragraph of the findings. We hope this makes the interpretation of Figure 2 easier.

3. The data is available within the article. While it is essential for transparency to include all the theories examined, could the 94 theories used once, and the 22 theories used twice be moved to an appendix? This might help reduce distractions from the analysis of the other theories that were used multiple times. This is a minor suggestion, and I understand it may not be feasible to have an appendix. Thank you for this suggestion. We had, in fact, done this in an earlier draft of the paper. Our main reason for bringing the text back is to ensure that all 252 articles in the search are referenced in the manuscript reference list. We think that the 371 words in these two paragraphs are not unduly lengthy or distracting.

4. Besides the point raised in point 2 about adding a small definition of what theory-article pairs mean, the article is clear and well-written. Essentially, it argues that since Mike Stein's call for care-leaving research to incorporate theory and build a care-leaving theory, researchers have made progress, but there is still much work to be done, particularly in developing care-leaving theory. In addition, the research produced an emerging theoretical framework (analytical typology) that helped analyse how the theory was applied in care-leaving research. This typology is especially valuable for those researching the care-leaving experience. Thank you for these positive comments: progress since Stein’s 2006 in the theorisation of care-leaving, more work and some fruitful avenues still lie ahead, and a potentially useful framework to assessing the extent to which a study is theoretically informed or driven.

---

## [Editor Report · Decision Letter 1]

21 May 2025

Theories used to explain care-leavers’ journey out of care: A scoping review

PONE-D-24-53785R1

Dear Dr. van Breda,

We’re pleased to inform you that your manuscript has been judged scientifically suitable for publication and will be formally accepted for publication once it meets all outstanding technical requirements.

Kind regards,

Daryl Higgins, PhD

Academic Editor

PLOS ONE
---

## [Editor Report · Acceptance letter]

PONE-D-24-53785R1

PLOS ONE

Dear Dr. van Breda,

I'm pleased to inform you that your manuscript has been deemed suitable for publication in PLOS ONE. Congratulations! Your manuscript is now being handed over to our production team.

Kind regards,

on behalf of

Professor Daryl Higgins

Academic Editor

PLOS ONE